# ParaGAN: A Cloud Training Framework for Generative Adversarial Networks

Ziji Shi*[†], Fuzhao Xue*, Jialin Li*, Yang You*
* National University of Singapore
[†] Alibaba Group

*Abstract*—**Generative Adversarial Network (GAN) has shown tremendous success in synthesizing realistic photos and videos in recent years. However, training GAN to convergence is still a challenging task that requires significant computing power and is subject to training instability. To address these challenges, we propose ParaGAN, a cloud training framework for GAN optimized from both system and numerical perspectives. To achieve this, ParaGAN implements a congestion-aware pipeline for latency hiding, hardware-aware layout transformation for improved accelerator utilization, and an asynchronous update scheme to optimize system performance. Additionally, from a numerical perspective, we introduce an asymmetric optimization policy to stabilize training. Our preliminary experiments show that ParaGAN reduces the training time of BigGAN from 15 days to just 14 hours on 1024 TPUs, achieving 91% scaling efficiency. Moreover, we demonstrate that ParaGAN enables the generation of unprecedented high-resolution ($1024 \times 1024$) images on BigGAN.**

## I. INTRODUCTION

Last decade has witnessed the success of Generative Adversarial Networks [7], which has a wide range of applications including image super resolution [13], image translation [8], [26], photo inpainting [6], [24]. However, training GAN at scale remains challenging because of the computational demands and optimization difficulties. Unlike Convolutional Neural Networks (CNN) or Transformer-based architectures where optimization is straightforward by taking gradient descents on *a single model*, there are *two sub-networks* to optimize in GAN, namely generator and discriminator. The generator samples from the noise and produces a fake sample as close to the real sample as possible, and the discriminator evaluates the generated sample. The generator aims to fool the discriminator, and the discriminator will try to identify the fake images from the real ones. Since the two components are optimized for two contradicting goals, it has been observed that GANs are difficult to converge. Therefore, to speed-up the GAN training at large-scale, we need a framework optimized on both system and numerical perspective.

Due to the difficulty of optimizing GAN, many state-of-the-art GAN models take days or even weeks to train. For instance, BigGAN [2] took 15 days for 8x V100 GPUs to train 150k steps. Table I summarizes the reported training time of some of the state-of-the-art GAN models. This has made it difficult to quickly reproduce, evaluate, and iterate GAN experiments. Also, current GAN frameworks usually support training with very few nodes.

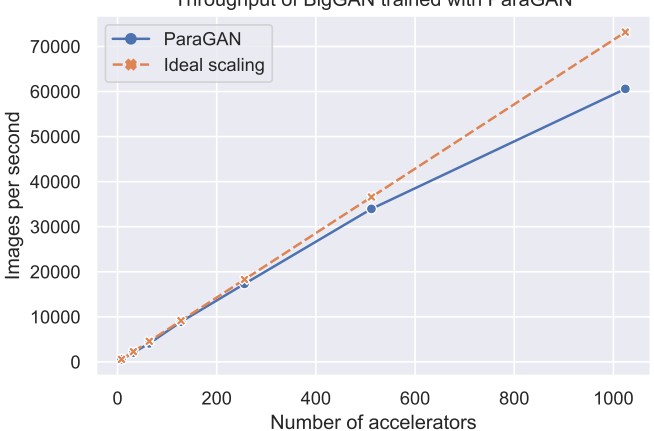

Fig. 1: ParaGAN scales to 1024 TPU accelerators at 91% scaling efficiency.

TABLE I: Training Time and Parameters Number for GANs trained on ImageNet 2012 dataset.

| Model | Accelerator | Training Time | # Params |
|---|---|---|---|
| SNGAN [17] | 8× V100 | 3d 13.6h | 81.44M |
| SAGAN [25] | 8× V100 | 10d 18.7h | 81.47M |
| BigGAN [2] | 8× V100 | 15 d | 158.42M |
| ContraGAN [10] | 8× V100 | 5d 3.5h | 160.78M |
| ProgressiveGAN[1] [12] | 8× V100 | 4d | 43.2M |

We argue that training speed is an important yet often ignored factor in the current GAN training landscape, and we propose to accelerate it with distributed training. But distributed GAN training has several challenges. First of all, most data centers have storage nodes and compute nodes separated for elasticity, but network congestion can happen from time to time, which prolongs the latency between nodes and affects training throughput. Secondly, there are usually different types of accelerators in the data center, but each of them has unique optimal hardware characteristics. If ignored, it can lead to the under-utilization of accelerators. Last but not least, training GAN at scale may cause a convergence problem, in which the GAN loss does not converge to a stable equilibrium. Therefore, this framework has to consider both system and numerical perspectives.

In this work, we present **ParaGAN**, a distributed training framework that supports large-scale distributed training for high-resolution GAN. We identify the performance bottlenecks

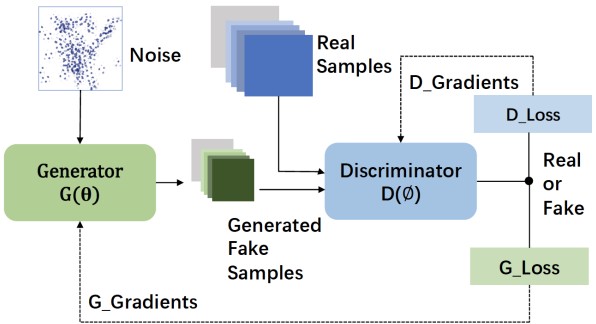

Fig. 2: Typical GAN architecture.

when training at scale and optimize them for efficiency. To stabilize the training process, ParaGAN comes up with an asynchronous update scheme and asymmetric optimization policy. ParaGAN has a simple interface for building new GAN architecture, and it supports CPU, GPU, and TPU.

The main contributions of ParaGAN include:

- We design and implement a scalable distributed training framework for GAN with optimizations on both system and numerical perspectives. With ParaGAN, the training time of BigGAN can be shortened from 15 days to 14 hours with 1024 TPU accelerators at 91% scaling efficiency, as shown in Fig. 1. ParaGAN also enables direct photo-realistic image generation at unprecedented $1024 \times 1024$ resolution, which is $4\times$ higher than the original BigGAN model.
- From the system perspective, we use a congestion-aware data pipeline and hardware-aware layout transformation to improve the accelerator utilization.
- From the numerical perspective, to improve the convergence for distributed GAN training, we present an asynchronous update scheme and asymmetric optimization policy.

## II. Background

As shown in Fig. 2, a GAN consists of a generator and a discriminator. The generator generates fake data samples, while the discriminator distinguishes between the generated samples and real samples as accurately as possible. The learning problem of GANs is a minimax optimization problem. The goal of the optimization is to reach an equilibrium for a two players problem:

$$\min_{G} \max_{D} \mathbb{E}_{x \sim q_{data}(x)} \left[ \log D(x) \right] + \mathbb{E}_{z \sim p(z)} \left[ \log \left( 1 - D(G(z)) \right) \right]$$

where $z \in \mathbb{R}^{d_z}$ is a latent variable drawn from distribution $p(z)$. The discriminator seeks to maximize the sum of the log probability of correctly predicting the real and fake samples, while the generator tries to minimize it instead. Formally, the convergence of GAN is defined as a type of Nash Equilibrium: one network does not change its loss regardless of what the other network does.

Since the two networks have contradicting goals, the training process of GAN is a zero-sum game and can be very unstable. Recent works show that i) GAN may converge to points that are not local minimax using gradient descent, in particular for a non-convex game which is common [5], [9], and ii) gradient descent on GAN exhibits strong rotation around fixed points, which requires using very small learning rates [1], [16]. Also, GANs training is sensitive to the hyperparameters and initialization [15]. Therefore, it is observed that GANs are difficult to optimize, and this is also the reason why it takes a long time to train them.

There are some existing GAN libraries [4], [11], [14], [15] to train state-of-the-art GANs. They provide standardized building blocks like network backbone and evaluation metrics, making it easy to build new models. However, they focus less on the system performance, and training GAN still takes days if not weeks. [18] benchmarks the performance of the various GANs within different network-related applications, [3], [22], [23] propose GAN-optimized hardware architectures. Different from the prior works, we aim to build a system that can be run on the public cloud using commodity accelerators. If the training process can be massively paralleled, the GAN community will benefit from it.

In ParaGAN, we adopt a co-designed approach: on the system level, we identify that the performance bottlenecks are rooted in network congestion and low accelerator utilization when training on the cloud, and ParaGAN implements a congestion-aware data pipeline and hardware-aware layout transformation to mitigate the issues; on the optimization level, we observe that it is beneficial to decouple the training of generator and discriminator, and ParaGAN proposes an asynchronous update scheme and an asymmetric optimization policy.

## III. Design and Prototypical Implementations

In this section, we will give an overview and discuss the design decisions of ParaGAN. We recognize that, the scalability is usually limited by the latency between nodes. Furthermore, when scaling up the GAN training, the numerical instability problem happens more often. We divide the following discussions into two folds and present our co-designed approach for system throughput and training stability.

### A. Programming Model

The design of ParaGAN is presented in Fig. 3. ParaGAN (blue region) is implemented on top of TensorFlow (green region) because TensorFlow provides the low-level APIs for model checkpointing, evaluation, and visualization. Different from TensorFlow, we provide high-level APIs for GAN which includes scaling manager, evaluation metrics, and common network backbones. Users of ParaGAN can import from ParaGAN or define their own components. ParaGAN then performs layout transformation and invokes TensorFlow, which converts the model definition into a computational graph. An optional XLA [20] pass can be performed followed by that. After that, the training starts on the CPU host and accelerators.

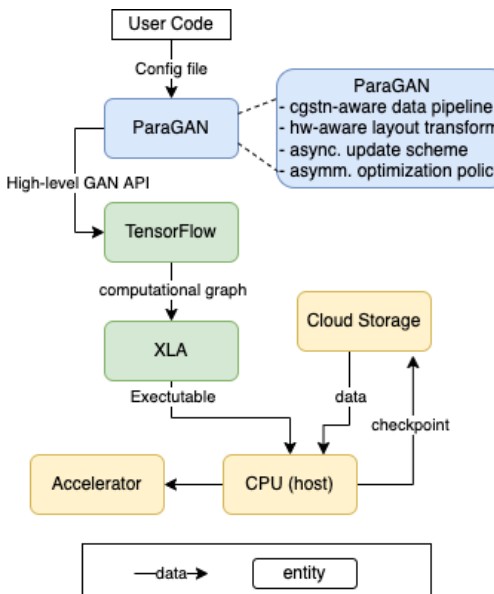

Fig. 3: Overview of ParaGAN architecture.

Listing 1: Inferface of ParaGAN

```python
import paragan as pg

class Generator:
  def model_fn(latent_var, y):
      # generator model
      return output

class Discriminator:
  def model_fn(x, y):
      # discriminator model
      return output, out_logit

scale_mgr = pg.ScalingManager(config=cfg,
                bs=2048, num_workers=128)

g = Generator()
d = Discriminator()
gan = pg.Estimator(g,d)

# train
for step in cfg.max_steps:
  scale_mgr.train(gan)

# evaluate
scale_mgr.eval(metric='fid')
```

We introduce a few concepts in ParaGAN:

*1) Scaling Manager:* The scaling manager is responsible for tuning the hyper-parameters that need adjustment during scaling. Users can start with the best hyper-parameters from a single worker and ParaGAN will properly scale them based on the number of workers based on the heuristics (eg. linear scaling, cosine scaling).

*2) Network Backbones:* It is common that one starts by building upon existing GAN architectures. We also provide some popular GAN architectures as backbone, including but not limited to:

- BigGAN [2];
- Deep Convolutional GAN (DCGAN) [19];
- Spectral Norm GAN (SNGAN) [17]

*3) Evaluation Metrics:* Evaluation metrics can be implemented differently across papers, and this can cause inconsistency. We provide commonly used evaluation metrics including Frechet Inception Distance (FID) and Inception Score (IS).

### B. System Optimizations

To satisfy the scalability requirement, we design ParaGAN with optimizations on I/O and computation.

We optimize the I/O performance by building a congestion-aware data pipeline. For data centers, the compute and storage nodes are usually interconnected via Ethernet instead of high-speed InfiniBand. The network traffic between them is not always stable since the infrastructure is shared with other tenants. This could cause problems when the training scales since latency fluctuates when the number of workers increases. Therefore, we implement a congestion-aware data pipeline to reduce the impact of network jittering.

To achieve a higher accelerator utilization, we perform hardware-aware layout transformation. A data center usually has multiple types of accelerators, and different accelerators have different architectures and preferred data layouts. For example, Nvidia A100 GPUs prefer half-precision data in multiples of 64 and single-precision data in multiples of 32, while previous generations prefer $8\times$. For TPU v3, the preferred data dimension should be a multiple of 128. Using the preferred data layout can increase the accelerator utilization, but it is usually up to the user to determine it. We come up with a hardware-aware layout transformation to transform the data into an accelerator-friendly format to maximize accelerator utilization.

### C. Numerical Optimizations

One of the main contributions of ParaGAN is its use of asymmetric training to improve the stability of GAN. As the number of workers increases, a larger batch size can be used to speed up the training process. However, we have noticed that the performance of large batch training for GAN is often unstable, and mode collapse occurs frequently. This issue arises because mode collapse is a type of GAN failure that occurs due to a highly coupled optimization process. To address this problem, ParaGAN introduces an asymmetric optimization policy and asynchronous update scheme, which help to decouple the optimization process and prevent mode collapse.

### IV. Implementation

To start, we conducted a profiling of BigGAN training using native TensorFlow [15] and the results are shown in Figure

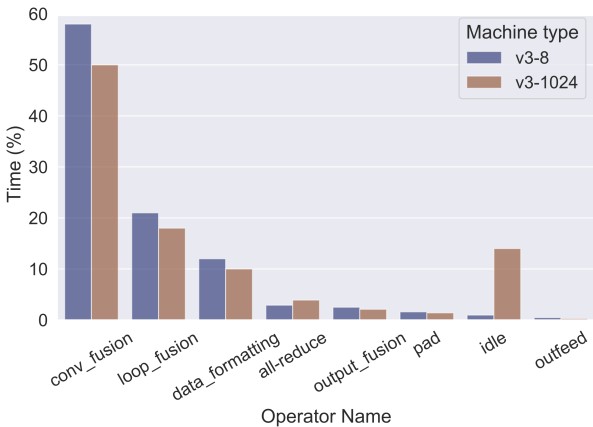

Fig. 4: Operator usage profile when training at scale.

4. As we scaled up the cluster size from 8 to 1024 TPU workers, we observed a significant increase in idle time due to the higher communication overhead. Nevertheless, convolution operations continued to take up the majority of the execution time, which suggests that training GAN is a compute-bound task. Therefore, our focus for achieving scalability in Para-GAN is on maximizing the utilization of accelerators.

To achieve this goal, we use congestion-aware data pipelining to reduce data pipeline latency, hardware-aware layout transformation to increase accelerator utilization, and mixed-precision training with bfloat16 for reduced memory.

### A. Congestion-Aware Data Pipelining

Network jittering can have a significant impact on training throughput due to the gradient synchronization stage, where all workers synchronize the gradient at the end of each step, and the time taken to complete this step depends on the slowest worker.

Although both TensorFlow and PyTorch implements data pipelines to hide the data loading latency, when severe network jittering happens, data loading and pre-processing takes much longer than usual, and it can be a bottleneck in large-scale distributed training. As shown in Fig. 4, when the number of workers scales from 8 to 1024, it spends 13.6% more time on idling, while data outfeeding time stays close. This indicates that the accelerators are busy waiting for data infeed and gradient synchronization, which leads to reduced utilization.

ParaGAN dynamically adjusts the number of processes and pre-processing buffer size based on the high variance network. It achieves this by using a sliding window to monitor network latency during runtime. If the current latency exceeds the threshold $\lambda$ over the window, the system increases the number of threads and buffer for pre-fetching and pre-processing. Once the latency falls below $\lambda$, the system releases the resources for pre-processing. This may result in an increase in shared memory usage, but shared memory is not typically a bottleneck and is often underutilized.

### B. Hardware-Aware Layout Transformation

Zero-padding is used in GAN when the input cannot fit into the specified convolution dimension. For example, a matrix of $100\times100$ will need 14 zeros padded around it to run on a $128\times128$ matrix unit. However, zero-padding hinders the accelerator performance because memory is wasted by padding, leading to a lower accelerator and memory utilization rate.

We implement ParaGAN by making sure both the batch size and feature dimensions are multiples of 128 whenever suitable. In NCHW (batch_size x number of channels x height x width) format, we implemented ParaGAN such that N/H/W are multiple of 128 on the host side so that the accelerator memory can be efficiently utilized.

On top of the feature dimensions, ParaGAN also seeks opportunities to batch data, in order to combine the intermediate result to be a multiple of optimal layout dimension without affecting the results. Such opportunities can be found at *reshape* and *matmul* operators. For instance, if two input matrices are to multiply the same weight, we can concatenate the two input matrices first before the matrix multiplication. In some senses, this is similar to operator fusion, but the key difference here is that ParaGAN's layout transformation is dependent on the hardware, so that the fused result can confine to the optimal layout.

### V. PRELIMINARY EVALUATION

In this section, we aim to answer the following questions: 1) how is the performance of ParaGAN compared to other frameworks? 2) how much does each part of the system contribute to the overall performance? And 3) what are the effects of the numerical optimizations on convergence?

In this section, we first evaluate the end-to-end performance of ParaGAN using three metrics:

- **steps per second** measures the number of steps ParaGAN can train per second;
- **images per second** measures the throughput of ParaGAN trained with ImageNet 2012 dataset;
- **time to solution** measures the time it takes to reach 150k steps on ImageNet at $128 \times 128$ resolution.

We first compare ParaGAN with other popular frameworks for end-to-end performance (Sec. V-B), and evaluate the scaling efficiency for ParaGAN (Sec. V-C).

### A. Experiment Setup

We choose BigGAN on ImageNet ILSVRC 2012 dataset as benchmark, because BigGAN has a profound impact on the high-resolution image generation, and it has a high computational requirement (Table I). On the other hand, ImageNet contains a good variety of classes (1000 classes), and it is usually challenging to train on. For the hardware backend, we first compare the performance of different backends, then we choose TPU due to accelerator availability reason.

While we use BigGAN to benchmark ParaGAN, our framework is generally applicable to other GAN architectures and dataset, and it is not tightly coupled with any specific accelerator backends.

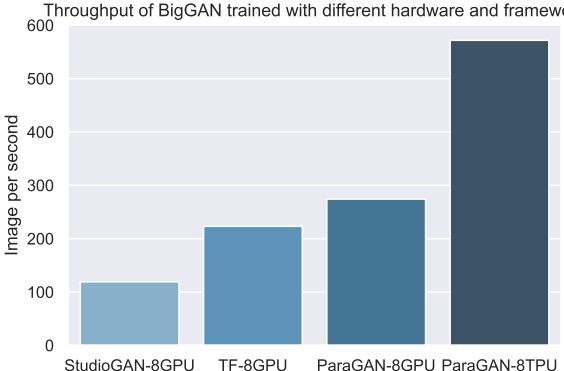

Fig. 5: Throughput of different systems and hardware combinations.

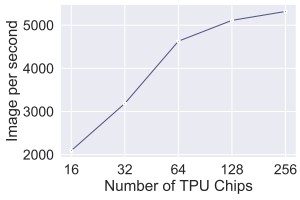
(a) Image per second.

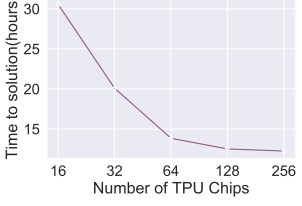
(b) Time-to-solution.

Fig. 6: Strong scaling with ParaGAN. Each TPU chip has two accelerators.

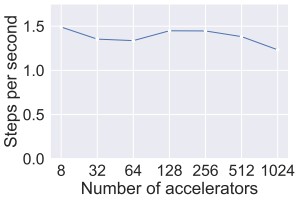
(a) Step per second.

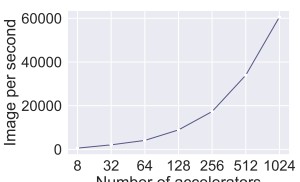
(b) Image per second.

Fig. 7: Weak scaling with ParaGAN.

## B. Framework-level Experiments

In Figure 5, we present a comparison of ParaGAN with StudioGAN [11] and native TensorFlow [15] in terms of GPU performance. In each experiment, we train BigGAN on ImageNet at a resolution of 128x128. We utilize eight Tesla V100 GPUs for all settings, except for ParaGAN-8TPU.

We observe that ParaGAN outperforms both the native TensorFlow and StudioGAN with 8 GPUs. We conjecture that the performance gain on the GPU setting mainly attributes to the use of congestion-aware data pipeline and hardware-aware layout transformations. We also observe that the performance gap is further pronounced when switching to the TPU as the accelerator. Due to availability reasons, the following sections mainly focus on the TPU as the accelerator.

## C. Scaling Experiments

We will discuss the strong and weak scaling results in this section. In the strong scaling experiments, we keep the total workload constant and vary the number of workers to examine the speedup on time-to-solution. Whereas in the weak scaling experiments, we keep the per worker workload (*batch size per worker*) constant and increase the number of workers.

*1) Strong Scaling:* For strong scaling experiments, we fix the total batch size to be 512 and train for 150k steps as target workload. Note that in order to be consistent with other experiments, we train on BigGAN at $128 \times 128$ resolution, with is smaller than the model trained in Fig. 1. We aim to study the effect of decreased per-worker workload when scaling.

As can be seen from Fig 6, with an increasing number of workers, the time to solution decreases from over 30 hours to 3 hours. We note that the scaling efficiency drops from 128 to 512 workers (64 to 256 TPU chips). This is because as we fix the global batch size to be 512, the per worker workload drops from 4 samples to 1 sample per batch, which under-utilizes the TPU. Thus, the time spent on communication overweights the computation when the batch size is too small. This is also verified by Fig 6, where the image per second barely improves with an increasing number of accelerator workers. However,

when the workload can saturate the accelerator, the scaling efficiency can be near optimal as shown in Fig. 1.

*2) Weak Scaling:* In the weak scaling experiments, we fixed the batch size per worker and evaluate the performance of our framework by increasing the number of workers. Firstly, we find the largest batch size for a single accelerator that does not lead to out-of-memory error. Then, we use the batch size for each worker, therefore, the amount of workload is kept identical across workers. The weak scaling experiments examine how well ParaGAN can handle communication with an increasing number of workers. As can be seen in Fig. 7, the trend in step-per-second is relatively steady even when using 1024 workers. It shows that ParaGAN can scale out well to a large number of workers while keeping a high scaling efficiency. It is worth noting that, as the number of workers scales, the system will be more likely to suffer from network jittering and congestion. A relatively flat curve (Fig. 7a) indicates that the data pipeline optimization in ParaGAN is effective in case of congestion.

## D. Accelerator Utilization

The basic computing unit of TPU is MXU (matrix-multiply unit), and a higher utilization is more desirable. We compare the accelerator utilization of BigGAN 128x128 on baseline [15] and ParaGAN. Fig. 8 shows that ParaGAN clearly outperforms native implementation with higher MXU utilization across different TPU configurations. We wish to highlight that even 2% improvements can be important when scaling to thousands of workers.

It is also worth noting that, with an increasing number of accelerators, the amount of communication increases, but ParaGAN is able to maintain a relatively higher utilization than native implementation, and the gap is increasing. It indicates

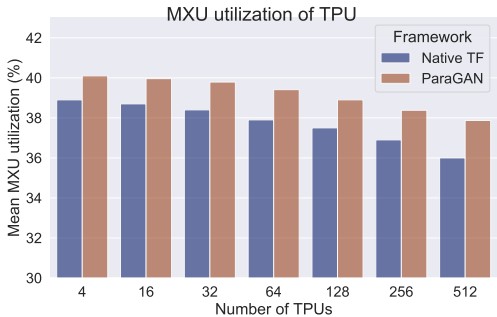

Fig. 8: Accelerator utilization of BigGAN trained with native TensorFlow and ParaGAN.

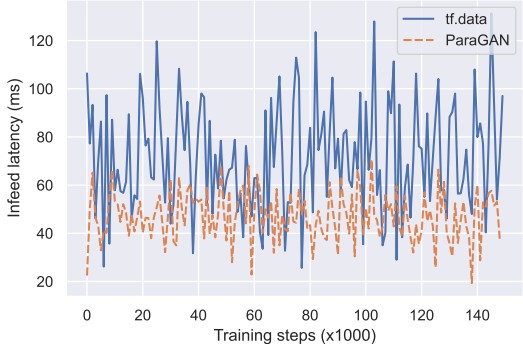

Fig. 9: Data pipeline latency.

that computation still dominates the training time as compared to native TensorFlow, and ParaGAN is able to keep up with scaling out.

**Data pipeline** provides 8-15% performance improvement over the baseline. When the number of accelerators increases, network jitter caused by congestion is more likely to happen, making data loading the slowest link in the training process. In ParaGAN, we try to saturate the accelerators by dynamically adjusting the buffer/CPU budget for the data pipeline. This is generally applicable, and ParaGAN enables this feature by default.

We compare the performance of our congestion-aware pipeline with TensorFlow's implementation. To ensure the results are comparable, they are run at the same time on the same type of machine with the same dataset directory, and latency is measured at the time taken to extract and transform a batch of data. As shown in Fig. 9, our pipeline tuner has a lower variance on latency.

**Layout transformation** and **operator fusion** combined provides 8% additional improvement by increasing the accelerator utilization. Considering that they both optimize on the kernel level, it is possible that we combine them into one pass by integrating layout-awareness into XLA. We also believe it may improve by using more aggressive layout transformations on intermediate result, but it might affect the convergence. We leave it as future work.

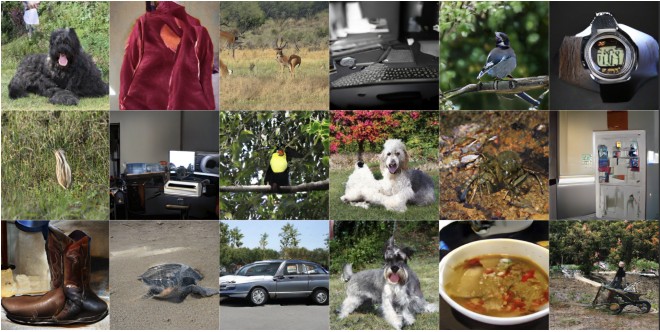

Fig. 10: Output of BigGAN at $1024 \times 1024$ resolution. Best viewed in colour.

### E. Generating High-Resolution Images

To our knowledge, we are the first to successfully train BigGAN at $1024 \times 1024$ resolution, which is $4\times$ larger than the original BigGAN. Training at high resolution is particularly hard, because generator will need to use more channels and deconvolutional layers to generate more details. It is therefore more sensitive to hyperparameters and initialization. Different from ProgressGAN [12] where they use progressive growing to train low resolution images first before increasing the resolution, we directly train it on $1024 \times 1024$ resolution, which is more challenging, and it requires the numerical optimization techniques we discussed.

The generated results achieves Inception Score (IS) [21] of 239.3 and Fréchet Inception Distance (FID) of 10.6. They are presented in Fig. 10 for visual evaluation.

### VI. DISCUSSION AND FUTURE WORK

ParaGAN is a large-scale distributed GAN training framework that supports high-resolution image generation with near-linear scalability. ParaGAN is optimized with an adaptive data pipeline, hardware-aware layout transformation, and an asynchronous update scheme for high throughput. To stabilize the training process of high-resolution GAN, ParaGAN also implements an asymmetric optimizer policy.

We hope ParaGAN will advance GAN research by accelerating the training process. ParaGAN scales almost optimally to 1024 accelerators, and it can greatly reduce the time to train a GAN model from weeks to hours. We leave it as future work to evaluate the performance of different GAN and diffusion model architectures on ParaGAN.

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
