# OpenReview forum: "ParaGAN: A Cloud Training Framework for Generative Adversarial Networks"
_iscaconf.org/ISCA/2023/Workshop/ASSYST — ASSYST Oral_

### Official Review · Reviewer_YzU5 · 2023-05-05
**Impressive results but could benefit from clarifying some of the technical contributions**

**Rating:** 6
**Confidence:** 4

**Review:**

I enjoyed the paper and found the improvements and the enablement of 1024X1024 resolution GAN training impressive. The scalability discussion would have benefitted from including the baseline TF scalability as well. Also in the evaluation,  relative effects of different techniques (data pipeline vs data layout vs asynchronous update) aren’t very clear. Some of this improvement is in the text, but it would be good to show these effects in a plot form.

The congestion-aware data pipelining idea seems to increase the host resources when it detects SLO violations in the network latency. My question here is could we not statically increase the amount of resources for the networking layer instead of the dynamic scheme proposed in the paper since the host typically is idle when the TPU is doing work? Additionally, I’m a bit confused how this technique is applicable to the TPU system since TPUs typically use ICI to communicate between each other?

Regarding the layout and data concatenation improvements for reshape and matmul operations, could/should this technique be implemented in the compiler instead?

**Review (Strengths/Weaknesses):**

Strengths:
* While GANs have recently been going out of fashion, this is still an important and interesting problem.
* Well written paper that motivates the problem well.
* Impressive improvement in both the training time and the fact that this technique enables the 1024X1024 resolution training.

Weaknesses:
* The evaluation lacks scalability studies of the baseline TF implementation. Without this, it is hard to gauge the improvement of ParaGAN over the baseline.
* The relative effects of different techniques aren’t well presented in the evaluation.
* The technical discussion could use more clarifications (see my questions above).


**Reviewer Expertise:**

Knowledgeable: I used to work in this area and/or I try to keep up with the literature but might not know the latest developments.

---

### Official Review · Reviewer_GNUV · 2023-05-08
**ParaGAN: A Cloud Training Framework for Generative Adversarial Networks**

**Rating:** 6
**Confidence:** 3

**Review:**

Summary:
* The authors present ParaGAN which is a framework built on top of Tensorflow to improve training time of Generative Adversarial Networks (GANs) and mitigate training instability.
* The paper proposed two main optimization, one is hardware aware layout to achieve high utilization of compute cluster, and second is an asynchronous optimization policy to combat numerical instability.
* The authors claim that the proposed framework allows 25x decrease in training time for BigGAN on 1024 TPUs

**Review (Strengths/Weaknesses):**

Strengths
* The paper performs a software characterization of GAN workload to identify appropriate optimization strategy.
* Interesting use if bfloat16 data type to mitigate network congestion problems
* The reported results depict significant improvements in training time.

Additional points to improve
* My biggest concern with the paper is that I does not detail how the proposed innovations, Congestion-Aware Data Pipelining and Layout transformations work.
* It is unclear how the framework interacts with code generation steps (eg XLA) when reacting dynamically with the system status (eg network congestion)


**Reviewer Expertise:**

Knowledgeable: I used to work in this area and/or I try to keep up with the literature but might not know the latest developments.

---

### Official Review · Reviewer_sEEX · 2023-05-12
**Well Motivated | Aims to Solve Practical Problem | Poor Experimentation and Results**

**Rating:** 5
**Confidence:** 3

**Review:**

The problem addressed is well motivated and is a practical limitation of GAN systems. The two game phases: Generator and Discriminator scale differently and there is also a sheer need for separation of data processing from retrieval. Especially, for disaggregated systems (such as in the cloud), where compute and storage are in different domains interconnected via network fabric the issues are more pronounced. Therefore, the authors address the correct issues in GAN. However, there is a plethora of literature addressing the above such as [1,3,4]. Addressing the challenges of deployment in cloud-scale is extremely vital, especially with different accelerators available such as GPUs, TPUs, and FPGAs (however, the authors miss evaluation with FPGAs). The authors propose congestion-aware data pipelining for throttling the resources depending on the load, which is critical for disaggregated systems but there is no concrete description of the mechanism neither conclusive empirical proofs via characterization. The experimentation procedure and results are neither comprehensive nor conclusive. The authors state the issue with the results but do not back it up with  empirical data . This diminishes the over-all merit as a cloud based evaluation would need conclusive proofs to identify the main bottlenecks.


**Review (Strengths/Weaknesses):**

Strengths:
1. The problem addressed is well motivated and is a practical limitation of GAN systems.
2. Different networks and characteristics of the layers, and addressing the decoupling of data processing from retrieval via efficient data pipelining for resource utilization as also identified by [1,3,4].
3. The throttling mechanism for congestion-aware data pipelining for issuing more threads (jobs) and managing the data-flow is a good idea.
4. Experimentation with GPUs and TPUs in the disaggregated cloud setup and characterizing the operator-by-operator impact is particularly a good contribution.

Weaknesses:
1. The problem addressed has been studied in great detail in literature [1,2,3,4]. However, the authors extend it to learn the impact in cloud based systems.
2. They evaluate their approach with GPUs and TPUs, however, they do not evaluate it based out of FPGAs which are also deployed for GAN systems and show performance benefits for end-to-end workflows as shown in [1.4]. In [4], the authors design the data-instruction pipelining by proposing hybrid MIMD-SIMD data processing in a FPGA platform, while decoupling data processing from retrieval to achieve performance benefits. ParaGAN should be evaluation with the design and platform such as [4].
3. The results and their interpretation is vague. For instance, the authors try to address the data jitter/delays across the network for IO and compute,  but do not show statistical results / proofs for the same. They simply state that their approach of congestion-aware data pipelining increases the performance, but there are no proofs that it is the contributor to performance gain or hardware optimizations.
4. The paper needs layer-by-layer characterization of the workload, and the major contributors to performance. The metrics chosen are vague and do not provide micro-architectural details for evaluating the benefits of ParaGAN.
5. As mentioned above, the experimentations are weak. The representation of the results is also poor and does not provide any vital information except for the fact that ParaGAN is more performant. We understand that the cloud may be "mystery box" but the results need to be replicated while providing the confidence interval to deterministically to justify the claims. The paper needs to provide conclusive proofs for their claim and identify the major bottlenecks in the system.
6. The authors should include the following relevant literature in the paper for covering the breadth.

References:
[1] Yazdanbakhsh, Amir, et al. "Ganax: A unified mimd-simd acceleration for generative adversarial networks." 2018 ACM/IEEE 45th annual international symposium on computer architecture (ISCA). IEEE, 2018.
[2] Navidan, Hojjat, et al. "Generative Adversarial Networks (GANs) in networking: A comprehensive survey & evaluation." Computer Networks 194 (2021): 108149.
[3] Chang, Jung-Woo, et al. "Towards design methodology of efficient fast algorithms for accelerating generative adversarial networks on FPGAs." 2020 25th Asia and South Pacific Design Automation Conference (ASP-DAC). IEEE, 2020.
[4] Yazdanbakhsh, Amir, et al. "Flexigan: An end-to-end solution for fpga acceleration of generative adversarial networks." 2018 IEEE 26th Annual International Symposium on Field-Programmable Custom Computing Machines (FCCM). IEEE, 2018.

**Reviewer Expertise:**

Knowledgeable: I used to work in this area and/or I try to keep up with the literature but might not know the latest developments.

---

### Official Review · Reviewer_9Yn6 · 2023-05-13
**Detailed Evaluation**

**Rating:** 6
**Confidence:** 3

**Review:**

This paper presents an approach to speed up the training of GANs. They propose a cloud training framework with certain optimizations such as hardware aware layout transformation, congestion-aware data pipeline, and asynchronous update.

Here are my comments:

1. According to figure 4, 13.6% more time in spent on idling in going from 8 to 1024 GPUs. How has this number improved with ParaGAN?

2. Is it really the case that shared memory usage is not a bottleneck? Even with larger batch sizes and large parameter models such as transformers? Can this be shown in a graph for the current setting?

3. In Figure 8, is the baseline system TPU for both the Native TF and ParaGAN?

4. Can you provide a cost comparison for the similar platforms?

**Review (Strengths/Weaknesses):**

Here are my comments:

1. According to figure 4, 13.6% more time in spent on idling in going from 8 to 1024 GPUs. How has this number improved with ParaGAN?

2. Is it really the case that shared memory usage is not a bottleneck? Even with larger batch sizes and large parameter models such as transformers? Can this be shown in a graph for the current setting?

3. In Figure 8, is the baseline system TPU for both the Native TF and ParaGAN?

4. Can you provide a cost comparison for the similar platforms?

**Reviewer Expertise:**

Knowledgeable: I used to work in this area and/or I try to keep up with the literature but might not know the latest developments.